# Urinary C3 levels associated with sepsis and acute kidney injury—A pilot study

**Sahra Pajenda[1], Florence Zawedde[1], Sebastian Kapps[1], Ludwig Wagner[1], Alice Schmidt[1], Wolfgang Winnicki[1], David O'Connell[2], Daniela Gerges[1]***

1 Division of Nephrology and Dialysis, Department of Internal Medicine III, Medical University of Vienna, Vienna, Austria, 2 School of Biomolecular and Biomedical Science, University College Dublin, Belfield, Dublin, Ireland

* daniela.knafl@meduniwien.ac.at

**Data Availability Statement:** All relevant data are within the paper and its Supporting Information files.

## Abstract

Acute kidney injury (AKI) is an abrupt deterioration of renal function often caused by severe clinical disease such as sepsis, and patients require intensive care. Acute-phase parameters for systemic inflammation are well established and used in routine clinical diagnosis, but no such parameters are known for AKI and inflammation at the local site of tissue damage, namely the nephron. Therefore, we sought to investigate complement factors C3a/C3 in urine and urinary sediment cells. After the development of a C3a/C3-specific mouse monoclonal antibody (3F7E2), urine excretion from ICU sepsis patients was examined by dot blot and immunoblotting. This C3a/C3 ELISA and a C3a ELISA were used to obtain quantitative data over 24 hours for 6 consecutive days. Urine sediment cells were analyzed for topology of expression. Patients with severe infections (n = 85) showed peak levels of C3a/C3 on the second day of ICU treatment. The majority (n = 59) showed C3a/C3 levels above 20 µg/ml at least once in the first 6 days after admission. C3a was detectable on all 6 days. Peak C3a/C3 levels correlated negatively with peak C-reactive protein (CRP) levels. No relationship was found between peak C3a/C3 with peak leukocyte count, age, or AKI stage. Analysis of urine sediment cells identified C3a/C3-producing epithelial cells with reticular staining patterns and cells with large-granular staining. Opsonized bacteria were detected in patients with urinary tract infections. In critically ill sepsis patients with AKI, urinary C3a/C3 inversely correlated with serum CRP. Whether urinary C3a/C3 has a protective function through autophagy, as previously shown for cisplatin exposure, or is a by-product of sepsis caused by pathogenic stimuli to the kidney must remain open in this study. However, our data suggest that C3a/C3 may function as an inverse acute-phase parameter that originates in the kidney and is detectable in urine.

## Introduction

The complement system with the central component C3 represents a member of the innate immune system. Three pathways of complement cascade activation have been characterized that converge at factor C3 and it is involved in immunological reactions in the kidney [1]. A

**Funding:** The author(s) received no specific funding for this work. The work was funded independently by the corresponding author.

**Competing interests:** The authors have declared that no competing interests exist.

most interesting example is its involvement in reperfusion injury [2], a serious form of acute kidney injury (AKI), through activation of the cascade and by influencing the regenerative potential through autophagy [3].

The most frequent cause of AKI in humans is infections such as sepsis [4, 5]. On a cellular level, the proximal tubular epithelium [6] represents the most vulnerable part of the nephron. Epithelial cells undergo damage at the brush border, loose cell polarity and are driven towards apoptosis [6, 7]. This occurs together with cast formation and tubular proteinuria. However, various cell types at the nephron possess compensatory mechanisms to counteract damage [8]. Thereby the tubular epithelial cells alter their gene transcription leading to changes in the regulation of autophagy [9]. This is of importance in order to balance and counteract stressing factors such as reactive oxygen species [10] which could finally result in AKI. Such regulations occur within various functional pathways [11] such as energy expenditure, inflammation, cell cycle [12, 13], lipid modification, metabolisms and autophagy [3]. As the result of such an ongoing process it has to be assumed that upregulated gene products, of which some represent secreted [13] or cleaved protein fragments [14, 15], might be detectable in urine and can serve as biomarkers [13, 16, 17]. In this line it is of note that at the kidney glomerular mesangial cells [18], glomerular epithelial cells [19] and tubular epithelia the complement factor C3 gene is induced for transcription upon pathogenic stimuli [2, 20, 21] given by interferon gamma (INFγ), immunocomplexes [22] and interleukin (IL)-1α [23]. C3 induction is also seen in kidneys of deceased organ donors [21]. It has been shown that this locally synthesized C3 is of relevance for complement-mediated injury of ischemic kidney allografts [2]. But this is just one side of the coin, the other side is that C3 has been shown to be involved in the regulation of autophagy in pancreatic beta cells which contributes to cell survival in inflammatory states [24] in pre-diabetic and diabetic conditions. This functional, cytoprotective mechanism of C3 has also been found to be of importance in limiting damage in renal epithelial cells upon exposure to toxic compounds such as cisplatin [25].

Motivated by the demonstration of the increasing importance of C3 in renal disease we have generated a monoclonal antibody recognizing a conformational structure of the anterior part of the C3a/C3 molecule [26]. We have selected a clone among others specifically recognizing native C3a/C3. Using this antibody, we followed patients admitted to the intensive care unit (ICU) because of serious infections for the secretion and appearance of C3a/C3 in urine. In addition, we analyzed the urinary sediment to visualize C3a/C3 distribution in various cellular structures. Concomitantly we sought to evaluate whether C3a/C3 is involved in urinary tract infections (UTI) often associated with such disease conditions.

## Materials and methods

### Sample collection

The study was approved by the ethics committee of the Medical University of Vienna under the EC number (721/2007). Eighty-five patients admitted to the ICU or intermediate care unit between 2009 and 2012 were included into this study after giving oral and written informed consent for participation. Only patients fulfilling at least 4 of 10 sepsis criteria [27] were selected and urine was collected from an indwelling urinary catheter every morning over the course of 6 days. Twenty one out of the 85 subjects were categorized as non-AKI patients. The interval between first and second time point of urine collection varied between 10 to 24 hours. Collection intervals 2, 3, 4, and 5 each represent a span of 24 hours.

The data on serum creatinine and other kidney function parameters as well as C-reactive protein (CRP) levels and leukocyte counts were extracted from the hospital data files. C3a/C3 urine analysis was performed in retrospect.

## Antibody generation

A C3a/C3 specific mouse monoclonal antibody (mAb 3F7E2) was produced and characterized as described earlier [26]. In brief, following three immunizations in two-week intervals, splenocytes were fused with the HAT sensitive murine myeloma cell line 536 using the polyethylene-glycol method as described earlier [28]. Outgrowing clones were screened by a multi-slot immunoblotting device (Merck Millipore, Bedford, MA, USA). Clones recognizing SDS treated C3 under non-reducing conditions were kept and further used for experiments.

## Dot blot

Thirty μl of urine were applied into an S&S Minifold I (Schleicher&Schüll, Germany) dot blotting device and filtered onto nitrocellulose by vacuum. The filter was then briefly dried and exposed to blocking solution for 30 mins. Following this the mAb 3F7E2 (tissue culture supernatant) was incubated over night at 4°C and next day the goat anti mouse HRP conjugated detection antibody (Dako, P0447) was incubated for 60 mins after two washing steps with TPBS. The filter was then developed by using a chemiluminescence reagent (Roche; BM Chemiluminescence Blotting Substrate, 11500694001) and recorded on an imaging device (Fusion Fx Vilber Lourmat) using Fusion software for signal detection. Pictures were further processed using Adobe Photoshop 6. Dot blot densitometric evaluation was performed using FusionCapt Advance Solo 4 16.06.

## Immunoblot

Human urine or human serum/plasma was loaded onto a 10% SDS-PAGE gel and run under non-reducing conditions, transferred onto nitrocellulose and incubated with tissue culture supernatant of mAb 3F7E2. The site of antibody binding was visualized such as described for the dot blot above.

## Urine C3a/C3 ELISA

Goat anti mouse pre-coated plates (Pierce) were washed once with PBS and the mAb 3F7E2 was bound onto the plate by incubating hybridoma cell supernatant for 2 hours. After one wash with TPBS, urine samples (100μl) were loaded together with a C3 standard series. Following an incubation period of 2 hours the ELISA plate was washed three times with TPBS (Tween20 PBS, 0,1%) using an ELISA washing machine. The rabbit anti human C3 Ab (Abcam, ab48342), conjugated with biotin, was incubated using Ray Bio-assay diluent (item $E_2$, 1:2500) for 1.5 hours under constant shaking. Following three washes with TPBS, the streptavidin HRP (Dako, Po397), diluted 1:2500 in Ray Bio-assay diluent, was bound to the biotin for 30 mins at room temperature under constant shaking. Following another three washes with TPBS, the two-component TMB peroxidase substrate chromogen solution (KPL, 50-65-00 and 50-76-01) was added and kept in the dark for 10 mins. The reaction was stopped with 1M HCl and read with the ELISA reader. Each value was calculated according to a standard curve.

**Urine C3a ELISA.** The human C3a ELISA (Invitrogen, BMS2089) was carried out as indicated in the test manual. Urine samples were thawed under airflow. The diluted (1:4 or 1:2 in sample diluent) urine samples (100μl) were pipetted into the prewashed test wells and incubated for two hours under constant shaking together with the provided standard series which was prepared in the sample diluent. Following washing on an automated ELISA washer (three times) the biotin-conjugate prepared in the assay buffer (100μl) was pipetted into each test well and incubated for one hour at RT under constant shaking. Following a second washing step

again three times, the streptavidin-HRP prepared in the assay buffer provided with the test kit (100μl) was added into each well and incubated for one hour under constant shaking at RT. After a final washing step, the TMB substrate solution (100μl) was pipetted into each well and reacted at RT for 20 minutes under light protection. The reaction was then stopped by adding 100μl stop solution. The test signal was read at 450nm at an ELISA reader and the sample concentrations were calculated according to the standard curve using the Gen5 version 2.03 program.

### Urinary sediment analysis

Seven ml of urine were centrifuged at 3000 RPM for 5 mins and the resultant sediment was resuspended in a tissue culture medium (RPMI 1640 containing 10% fetal bovine serum). Cytoslides were prepared using a Shandon cytocentrifuge and air dried for at least 2 hours. These were either wrapped in aluminum foil and frozen at -20°C or analyzed immediately. In brief, slides were fixed in acetone for 5 mins and the cell containing area was marked by drawing a cycle around the cells. Tissue culture supernatant of mAb 3F7E2 was then applied onto the slide and incubated over night at 4°C in a moist chamber. The next day the slides were washed in PBS and incubated in Alexa fluor 488 labelled donkey anti mouse secondary antibody for 1 hour at room temperature. Following two washing steps in PBS the slides were mounted in Vectashield (VECTOR, Z0619) mounting media containing propidium iodide for staining DNA and nuclei. Slides were viewed and recorded at a Zeiss confocal microscope or a Leica Aristoplan microscope. Pictures were further processed using Adobe Photoshop 6.

### Statistical analyses

Adherence to a Gaussian distribution was determined using the Kolmogorov-Smirnov test. Normally distributed data were described as means±SDs. The paired samples t-test was utilized to compare continuous variables within the same group. Qualitative variables were described with counts and percentages. The strength of association between CRP, leukocyte count, age, peak creatinine and urinary peak C3a/C3 was measured with the Pearson correlation coefficient. Data were analysed with SAS (version 9.2 for Windows) and SPSS (version 26 for Macintosh). All p-values result from 2-sided tests, with significance inferred at $p<0.05$.

## Results

Acute phase proteins in the serum are represented by CRP, fibrinogen, ferritin, complement factor C3 and others. These have their origin mainly in the liver. However, for the kidney such an acute phase reaction has not been described, but C3 production and secretion from various kidney epithelial cells has been demonstrated before [18, 19, 23, 29].

### Dot blot initial screens

In order to look for kidney born urinary proteins, which might be of diagnostic value in critical illness, previous literature indicated that some complement factors representing secreted proteins are induced by pathogenic stimuli such as infections. This motivated us to perform pilot testing using a dot blot analysis and the C3a/C3-specific monoclonal antibody 3F7E2. Using a 96-well dot blotting device a transient elevation of C3 secretion in urine was found in patients suffering from serious infection (**Fig 1**). Most patients exhibited a decline in urinary C3a/C3 secretion before transfer to the open ward (left panel of **Fig 1**), however, an increase of urinary C3a/C3 on days 5 and 6 could also be found and was in some individuals associated with worsening disease (right panel in **Fig 1**) and a fatal outcome (patient **o** in **Fig 1**, *Table 1*).

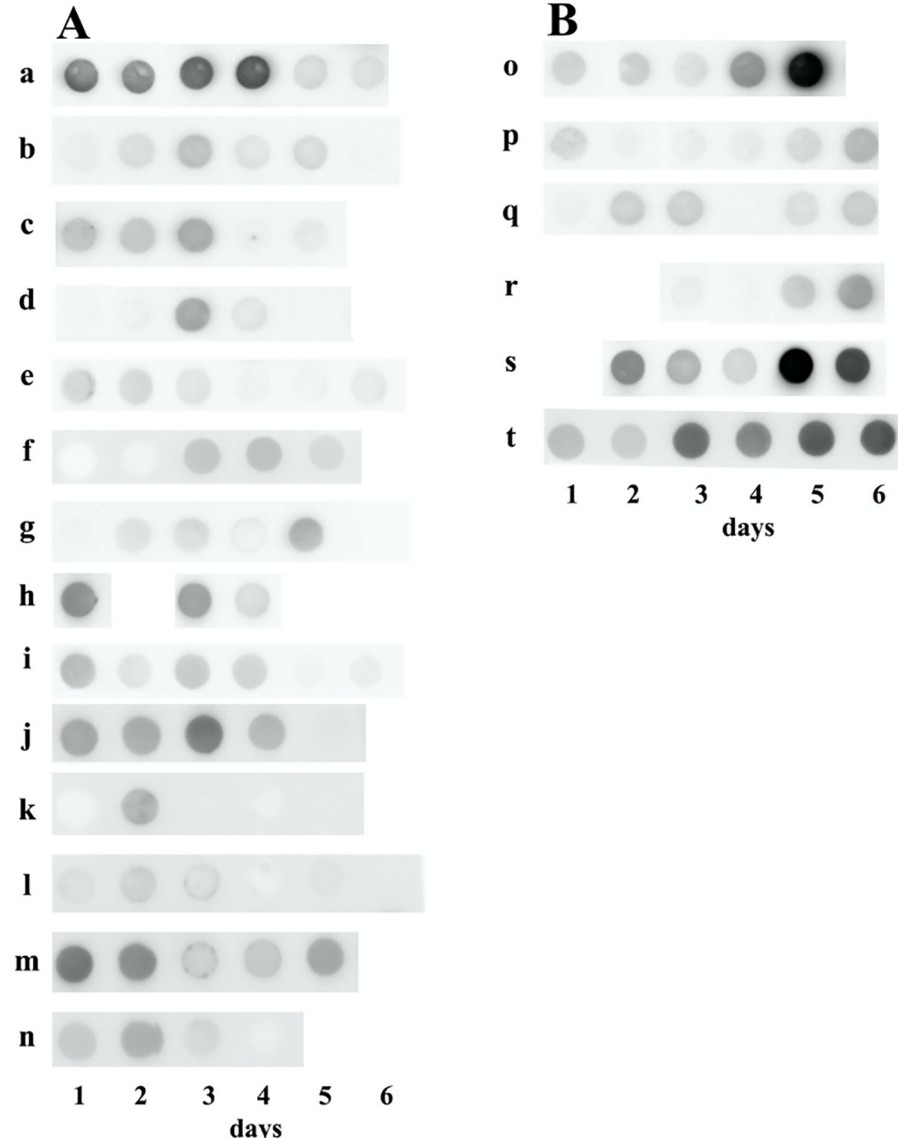

**Fig 1. C3a/C3 dot blot.** Urine samples obtained at consecutive time points from ICU patients suffering from serious infections. Patient (**o**) did not survive this episode due to multiorgan failure the same day as urine sample #5 was collected. Shown is a representative experiment out of four. Left panel of the figure represents patients with decreasing urinary C3a/C3 levels. Right panel shows patients with increasing urinary C3a/C3 levels towards 5[th] and 6[th] day of ICU treatment.

## ELISA testing

As a next step, we investigated if urinary C3a/C3 secretion correlated with other markers of systemic inflammation or stages of AKI. Therefore, urine from 85 sepsis patients (*Table 2*) undergoing ICU treatment was screened for C3a/C3 levels over a period of 6 days. As depicted in the left panel of **Fig 2** the highest mean overall level of C3a/C3 was found on the second day of measurement, which represented the second day of treatment (all p<0.05). Selectively measured C3a was detectable in urine over all 6 days (**Fig 2** right panel). Out of 230 samples 178

**Table 1. Densitometric values of C3/C3a dot blot analysis of 20 AKI patients at the ICU.** Urine samples were obtained at consecutive time points over 6 days.

| ID | Day 1 | Day 2 | Day 3 | Day 4 | Day 5 | Day 6 |
|---|---|---|---|---|---|---|
| a | 307 | 260 | 355 | 411 | 100 | 91 |
| b | 56 | 98 | 165 | 79 | 97 | 38 |
| c | 167 | 154 | 223 | 52 | 61 | na |
| d | 36 | 44 | 222 | 74 | 38 | na |
| e | 105 | 97 | 74 | 41 | 39 | 58 |
| f | 34 | 50 | 158 | 169 | 109 | na |
| g | 40 | 78 | 93 | 50 | 184 | na |
| h | 294 | na | 239 | 95 | na | na |
| i | 164 | 70 | 130 | 107 | 36 | 47 |
| j | 248 | 239 | 370 | 198 | 85 | na |
| k | 48 | 207 | 78 | 63 | 76 | na |
| l | 104 | 138 | 115 | 68 | 90 | 77 |
| m | 372 | 327 | 162 | 168 | 249 | na |
| n | 161 | 238 | 123 | 69 | na | na |
| o | 112 | 113 | 85 | 297 | 724 | na |
| p | 108 | 49 | 52 | 59 | 113 | 200 |
| q | 51 | 134 | 140 | 45 | 97 | 144 |
| r | na | na | 54 | 40 | 141 | 267 |
| s | na | 302 | 197 | 116 | 686 | 490 |
| t | 186 | 150 | 398 | 327 | 436 | 438 |

na = not applicable.

**Table 2. Demographics and disease status.**

|  | ICU (n = 85) |
|---|---|
| Age in y (mean ± SD) | 58.88 ± 15.02 |
| Age in y (median; min, max) | 62 (15, 86) |
| Male/Female | 49/36 |
| Pneumonia n (%) | 49 (57.6%) |
| UTI n (%) | 6 (7.1%) |
| CPR n (%) | 16 (18.8%) |
| MCI n (%) | 8 (9.4%) |
| TX n (%) | 10 (11.8%) |
| sCr (mean ± SD; mg/dL) | 1.87 ± 1.66 |
| CRP (mean ± SD; mg/dL) | 18.00 ± 11.24 |
| no AKI n (%) | 21 (24.7%) |
| AKI stage 1 n (%) | 37 (43.5%) |
| AKI stage 2 n (%) | 10 (11.8%) |
| AKI stage 3 n (%) | 17 (20.0%) |
| AKI on CKD n (%) | 28 (32.9%) |
| CKD without AKI (%) | 1 (1.2%) |

AKI–acute kidney injury; CKD–chronic kidney disease; CPR–cardiopulmonary resuscitation; CRP–C-reactive protein; MCI–myocardial infarction; sCr–serum creatinine; SD-standard deviation; TX–history of organ transplantation; UTI–urinary tract infection.

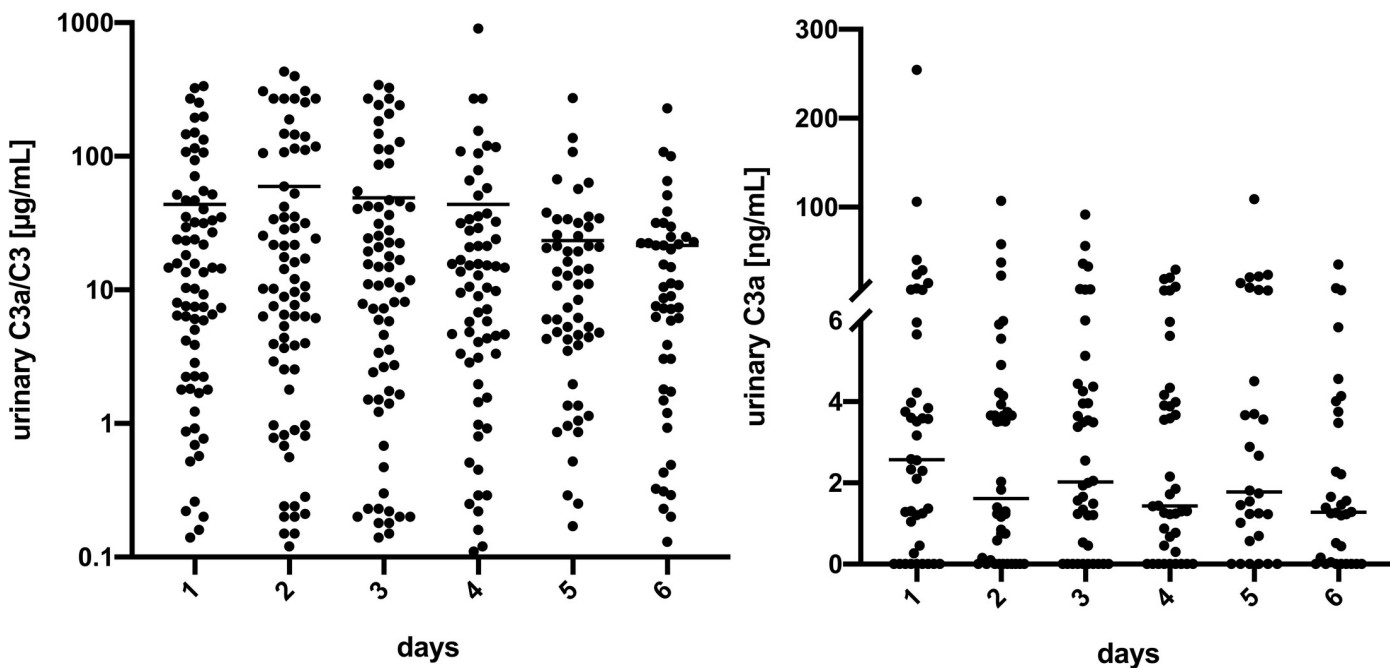

**Fig 2. ELISA-measurement of urinary complement factor C3a/C3 and C3a in 85 ICU patients.** Data are presented in columned scatter graphs; the horizontal lines mark the mean values of the respective C3a/C3 (left) and C3a (right) levels. Day 1: n = 78; day 2: n = 78; day 3: n = 72; day 4: n = 66; day 5: n = 56; day 6: n = 48.

exhibited detectable levels of C3a as shown in (**Fig 2** right panel). Thirty-seven patients had AKI stage 1, ten patients had AKI stage 2, and 17 had AKI stage 3. Twenty-one ICU patients with sepsis showed no signs of AKI (*Table 2*). The measured urinary C3a/C3 level did not correlate with AKI stage (**Fig 3**) and did not allow any conclusion regarding disease outcome.

There were 29 patients with a pre-existent chronic renal disease, out of whom 28 underwent an additional AKI during the infection period.

Most of the 85 patients were undergoing a remarkable acute phase reaction with a mean peak level of CRP of 18.00 mg/dl (±11.24) and a mean peak leukocyte count of 12.613 G/l (±6.746). Sixteen patients did not recover and died at the ICU.

Urinary C3a/C3 levels correlated negatively with serum CRP (p<0.0001). There was no correlation with peak serum creatinine, peak leukocyte count and age (*Table 3*).

## Immunoblot

In order to evaluate of what biological relevance this urinary C3a/C3 secretion might be, we sought to discover whether C3a/C3 is present in its entire molecular structure or only in fragments. For this reason, we loaded high C3a/C3 containing urine in SDS loading buffer onto SDS-PAGE gels. The blotted samples were developed with mAb 3F7E2. As demonstrated in **Fig 4** the entire 190 kDa C3 molecule was detected in the urine samples. For comparison the peripheral blood derived C3 is shown aside the immunoblot result from urine (**Fig 4**, middle panel). In addition, urine was concentrated up to 20 times using protein concentrators and the resultant urine concentrate was similarly immunoblotted as described above. C3 fragmentation and the C3a fragment were detected in lane 3 of **Fig 5**.

## Immunofluorescence

As a next step it was considered important to evaluate the origin of the urinary C3a/C3. Therefore, we stained urinary sediment for C3a/C3 expression obtained from 12 patients with

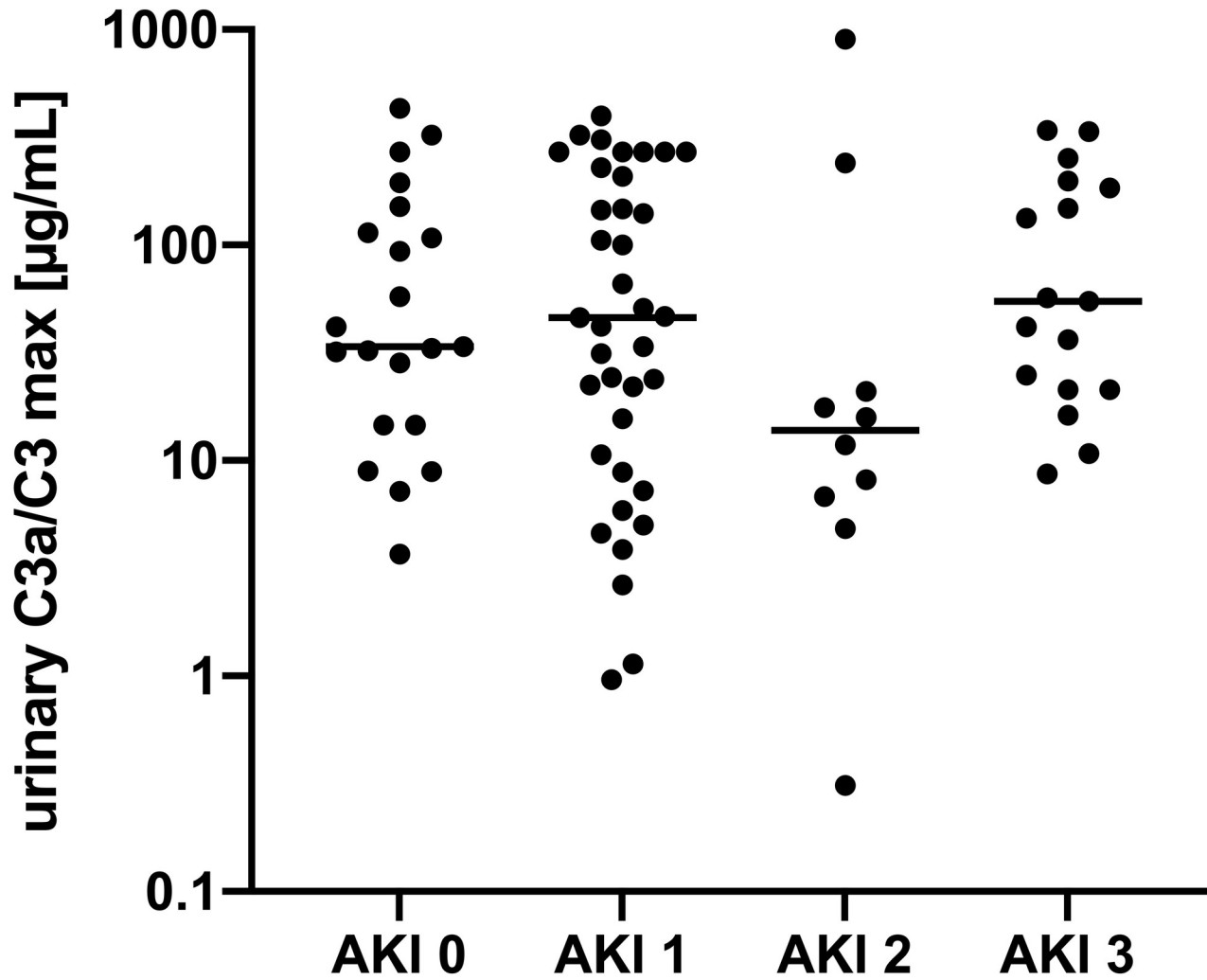

**Fig 3. Peak urinary C3a/C3 level in patients categorized for stage of AKI.** Peak levels of urinary C3a/C3 did not correlate with AKI stage. AKI 0 (n = 21), AKI 1 (n = 37), AKI 2 (n = 10) AKI 3 (n = 17).

serious infections. As depicted in **Fig 6A and 6B** intracellular granules of C3a/C3 could be visualized in cells with epithelial morphology. The staining pattern was highly variable, with large granular and vesicular staining indicating an autophagic granular expression (90% of positive cells, **Fig 6A**), by contrast, reticular cytoplasmic distribution indicated endoplasmatic reticulum production **Fig 6B**. This mode of positive staining was mainly restricted to epithelial cells. In addition, bacteria were found enveloped in a C3a/C3 detectable coat (**Fig 6C**). A negative control from a patient with non-inflammatory condition is included in **Fig 6D**.

In addition to the urinary sediment, human renal tissue from a patient with tumor nephrectomy showed areas of tubular epithelial staining as demonstrated in **Fig 7**.

## Discussion

In this study, we sought to investigate a urinary secreted factor indicative for stressing renal conditions. In this respect urine of 85 ICU patients, all admitted because of a serious infection and critical illness, was analysed. Sixty percent of these patients showed high secretion of C3a/ C3 (>20μg/ml peak level) mostly about 12–24 hours following admission. About 40% did not

**Table 3. Correlation of peak urinary C3a/C3 with age, CRP, WBC, peak sCr.**

|  | CRP | WBC | age | Peak sCR | Peak-uC3a/C3 |
|---|---|---|---|---|---|
| **CRP** | 1.00 | 0.20 | -0.04 | -0.11 | -0.41 |
|  | N/A | 0.061 | 0.74 | 0.33 | **<0.0001** |
| **WBC** | 0.20 | 1.00 | -0.11 | 0.02 | -0.08 |
|  | 0.06 | N/A | 0.30 | 0.87 | 0.46 |
| **Age** | -0.04 | -0.11 | 1.00 | 0.16 | -0.02 |
|  | 0.74 | 0.30 | N/A | 0.15 | 0.87 |
| **Peak sCr** | -0.11 | 0.02 | 0.16 | 1.00 | 0.09 |
|  | 0.33 | 0.87 | 0.15 | N/A | 0.39 |
| **Peak-uC3a/C3** | -0.41 | -0.08 | -0.02 | 0.09 | 1.00 |
|  | **<0.0001** | 0.46 | 0.87 | 0.39 | N/A |

Pearson correlation of CRP, WBC, log10 of peak serum creatinine during the 6 observation days (peak sCr) and log10 of peak urinary C3a/C3 (peak-uC3a/C3). The upper white line represents the correlation coefficient, the lower line in gray represents the p-value. A p-value of <0.05 was considered as statistically significant (bold). CRP–C-reactive protein; sCr–serum creatinine; WBC–white blood count.

present with such high urinary C3a/C3 levels, which did not appear to be associated with a milder course of disease compared to those presenting with high C3a/C3 levels. In addition, urinary C3a/C3 levels were not associated with AKI stage. Furthermore, we recorded peak CRP levels during the urine collection period. Interestingly, peak CRP levels correlated inversely with peak urinary C3a/C3. This unexpected observation might be due to a discordant gene regulation regarding C3 between the liver and the kidney tissue, mechanistically caused by different transcription enhancers and repressors in these functionally very different tissue types. A second explanation might be that complement activation and consumption in urine leads to lower C3 levels in some individuals. A third and interesting option is higher endophagocytosis (autophagy) in patients with higher CRP levels. Such results have already been demonstrated in beta cells of diabetic individuals [24]. Similarly to CRP, the peak leukocyte count was not associated with urinary C3a/C3. These data seem to indicate that C3 induction at the kidney is caused by stimuli other than those inducing CRP levels and leucocytosis in the peripheral blood during some forms of infection and sepsis.

Acute phase proteins are rapidly increased in transcription and translation upon inflammatory stimuli. The increase of these proteins in the serum is mainly orchestrated by the liver. However, it is of high interest whether other organs such as the kidney are also capable of producing following a pathogenic stimulus. For complement factors extrahepatic sites of synthesis have been researched for several years and their history is reviewed [29]. Interleukins such as INFγ, IL-1 and others have been effective in stimulating C3 production in *in vitro* studies although to a much lower extent when compared with that in the liver. The level of urinary C3 is much lower (in ng-μg/ml) than in serum (mg/ml) even under stress conditions. However, it is well demonstrated by earlier authors that proximal tubular cells can produce both C3 and factor B following IL-1α stimulation in a time and concentration dependent manner [23]. Local synthesis of C3 has already been described in the past. Sack S and colleagues were able to describe that glomerular mesangial cells are capable of producing C3 and C4, with an increase in C4 expression after stimulation with INFγ, whereas C3 expression remains unaffected under INFγ stimulation [18]. These data are substantiated by a further study, which demonstrates that C3 is synthesized, processed, and secreted by glomerular epithelial cells under basal conditions, with the C3 alpha and beta polypeptide chains having identical electrophoretic mobilities with those of hepatic C3. In contrast to the study mentioned above, stimulation

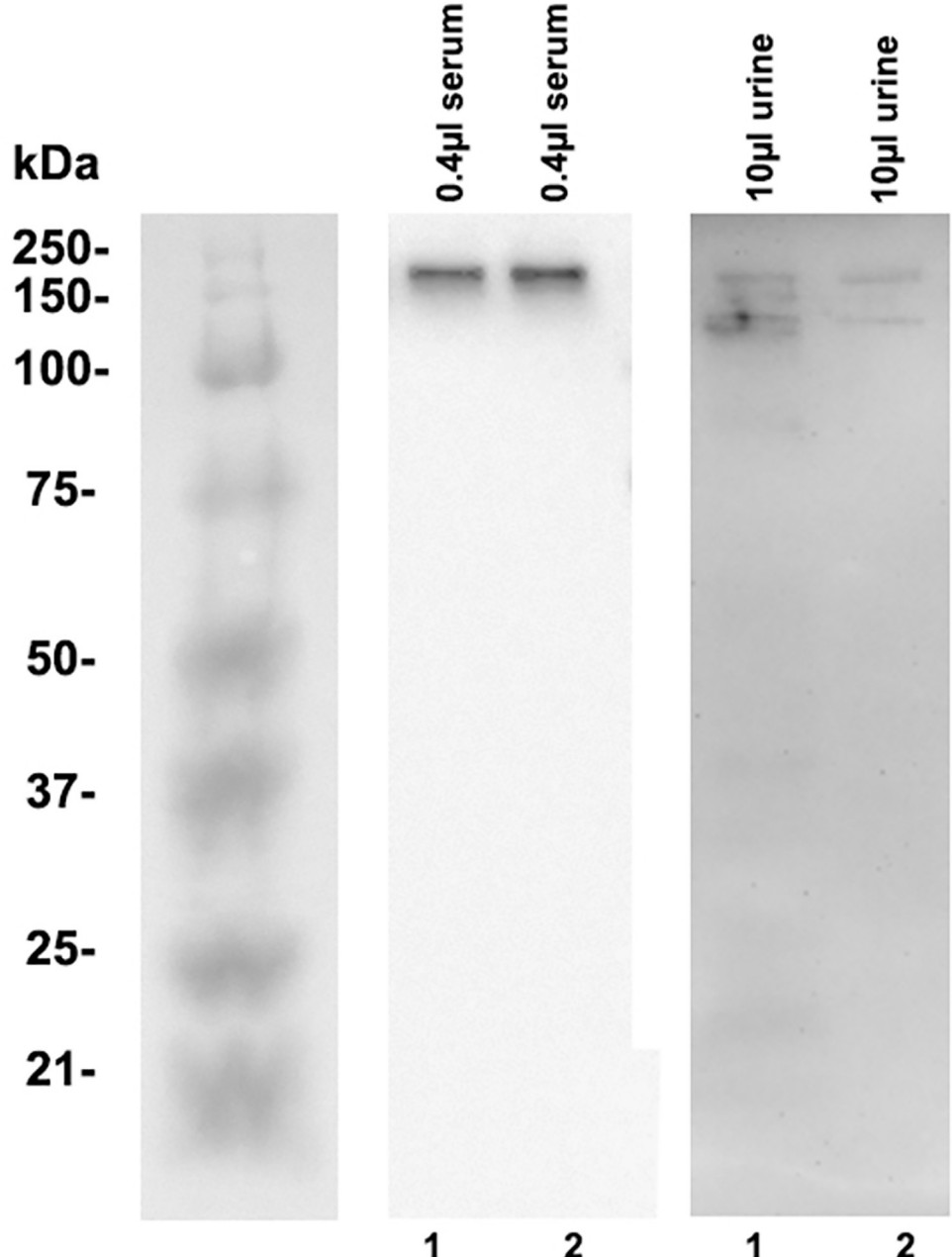

**Fig 4. C3 serum und urine immunoblot using mAb 3F7E2.** Ten µl of human urine (patient#1 and #2) was loaded onto a 10% SDS-PAGE transferred to nitrocellulose and incubated with mAb 3F7E2 (right panel). 0.4µl of human serum (patient#1 and #2) was loaded onto a 10% SDS-PAGE transferred to nitrocellulose and incubated with mAb 3F7E2 (middle panel). Molecular weight marker is shown at the left panel. Shown is a representative experiment out of two. The entire 190 kDa C3 molecule was detected and exhibited some extent of degradation. C3a could not be delineated.

with INFγ lead to an increase in C3 gene expression, indicating that C3 expression in glomerular epithelial cells is regulated by INFγ [19]. In addition, increased local C3 synthesis has been described in human diseases, such as postischemic acute renal failure and immune-mediated nephritis [2, 22].

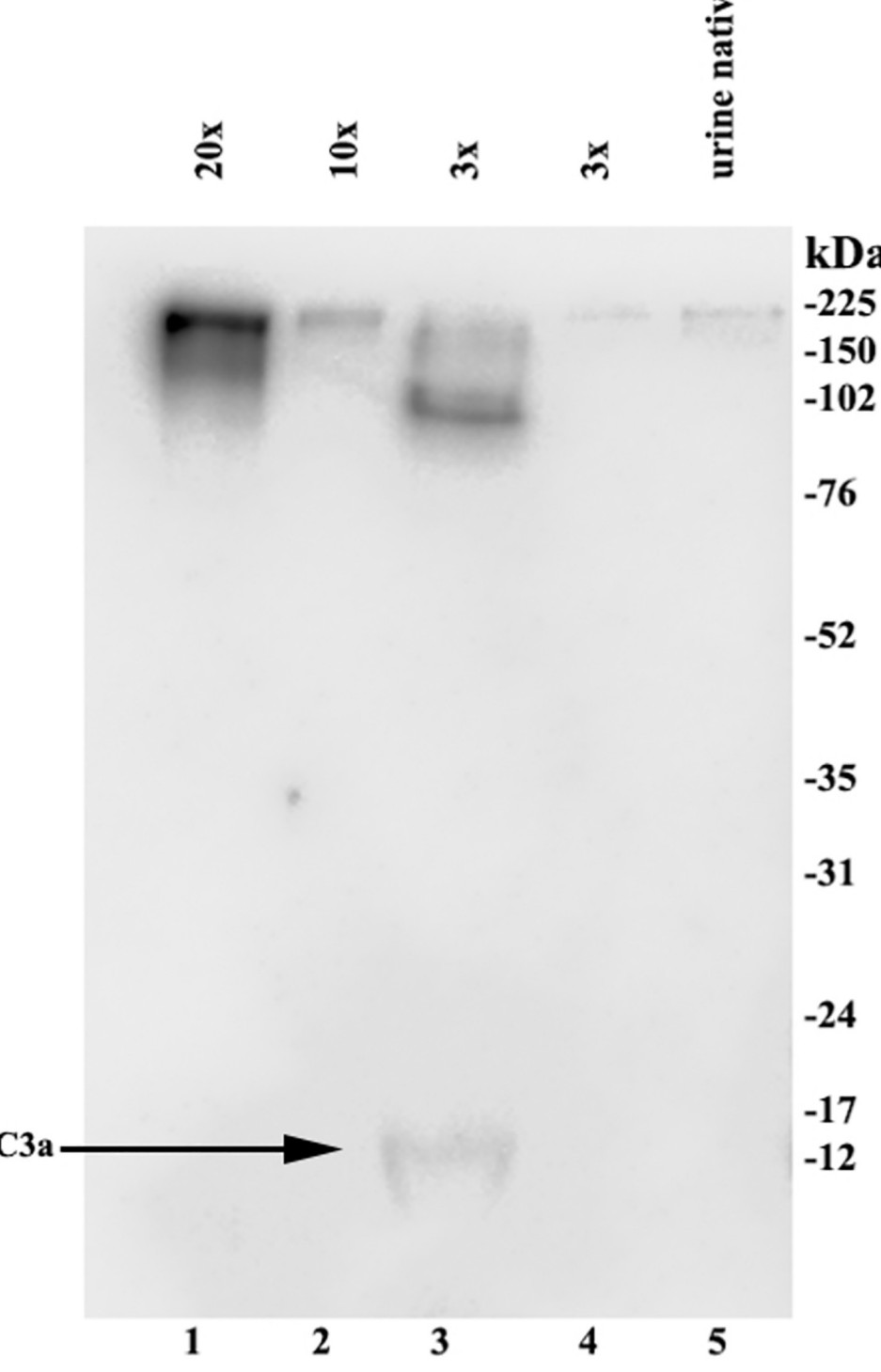

**Fig 5. C3a/C3 immunoblot developed with 3F7E2 mAb.** Twenty times (**20x**) concentrated urine from a patient with high C3a/C3 level in urine ELISA (lane 1); ten times (**10x**) concentrated urine from a patient with medium range C3a/C3 level in urine ELISA (lane 2); three times (**3x**) concentrated urine from patient with C3a fragment (shown by **arrow**) in urine (lane 3); three times (**3x**) concentrated urine from patient with undetectable C3a/C3 level in urine ELISA (lane 4); non-concentrated (**native urine)** urine (lane 5).

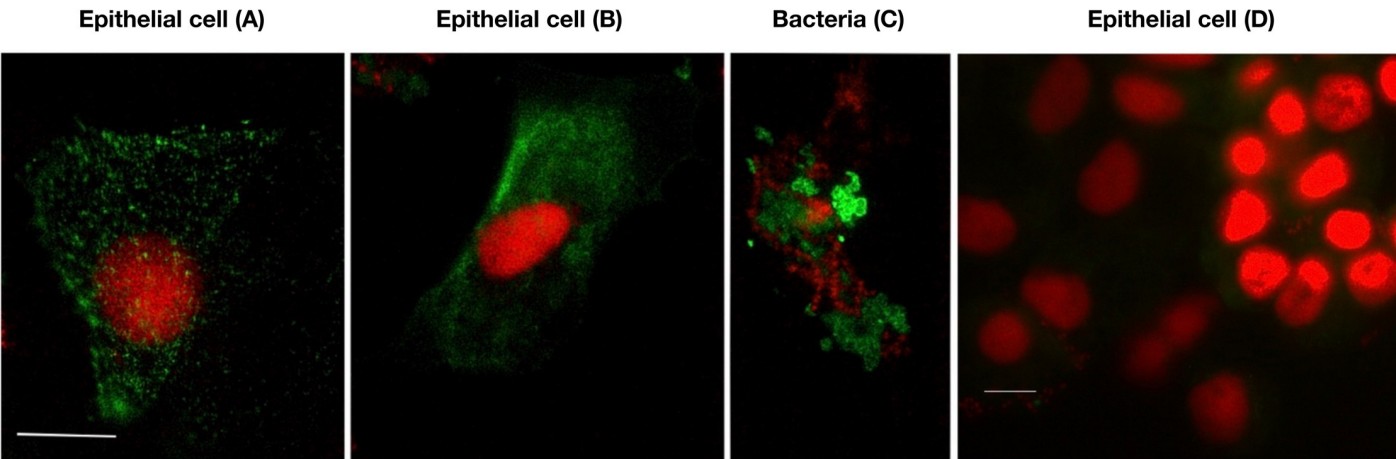

**Epithelial cell (A)** **Epithelial cell (B)** **Bacteria (C)** **Epithelial cell (D)**

**C3/Propidium iodide**

**Fig 6. Immunofluorescence staining of urinary sediment using mAb 3F7E2.** Tubular epithelial cells contain intracellular granules with C3 (**A, B**). Bacteria present in the urinary sediment are veiled in C3a/C3 to some extent (**C**). The scale bar represents 10 μm. Immunofluorescence staining of a urinary sediment of a patient without systemic inflammation and AKI served as control (**D**). The scale bar represents 10 μm.

The role of the complement cascade in AKI has been extensively reviewed by McCullough [30]. It is of particular note that the C3a receptor (C3aR) [31] and C5a receptor (C5aR) [32] are expressed at the tubular cell surface which is involved in upregulation of pro-inflammatory factors and chemokines to initiate granulocyte infiltration. In this respect, our article demonstrates for the first time to our knowledge, that tubular epithelial cells can secrete C3 upon infectious stimuli. Whether this C3 secretion in urine is of specific biological relevance or represents a by-product only must be left open by this study. However, as demonstrated in **Fig 6C** some bacteria are covered by C3a/C3 products. Therefore, it appears that C3 might also be involved in antibacterial defence and thereby relevant for opsonization and chemotaxis to attract macrophages and granulocytes to the site of infection.

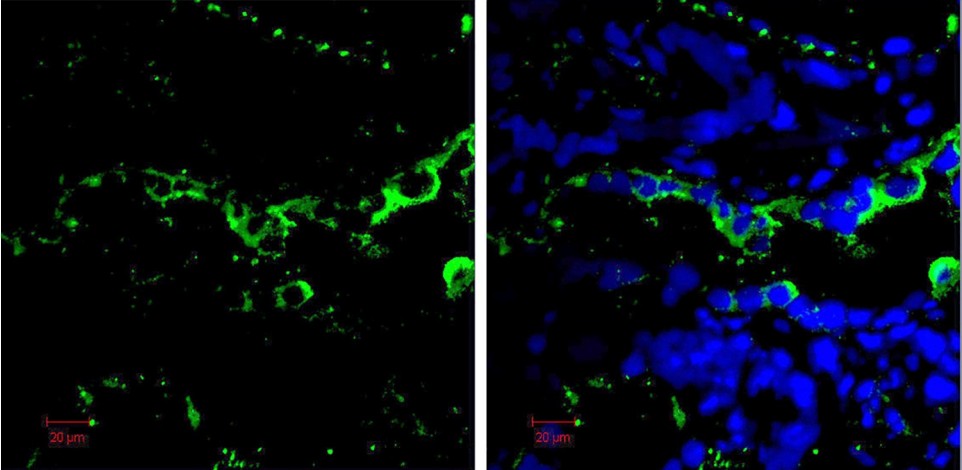

**Fig 7. Immunofluorescence staining of cryosections of a kidney removed because of a renal cell carcinoma using rabbit anti human C3-specific antibody.** Tubular epithelial cells contain intracellular C3 (left panel). Merging C3 staining with DAPI nuclear staining (right panel).

C3 is secreted from tubular epithelium and might represent a determinant of tubular epithelial stress. When this stress is elevated the transcription and secretion is increasing. We strongly suggest that urinary C3 of critically ill patients suffering from serious infections is the result of synthesis and secretion from renal epithelial cells. For this reason, we have excluded patients suffering from nephrotic syndrome to circumvent the chance of detecting blood born C3. Furthermore, we stained urinary born cells fixed on cytopreparation slides for presence of C3a/C3 in tubular epithelial cells. This demonstrated the various extent and morphologic distribution of intracellular C3 in renal tubular epithelial cells. Two modes of secretion are known in biology. It has to be assumed that C3 is not constitutively secreted into urine, but that there must be a regulated mode of secretion from nephronic epithelial cells. A specific stimulus secretion coupling appears to exist with one of them occurring in infection and sepsis. The exact triggering molecule has not been characterized yet but must differ from that in the liver. Transcription factor and repressor proteins are usually expressed in a tissue- and cell subtype-specific manner. Thereby, renal epithelial cells express different transcription factors and repressor proteins than hepatocytes. The function of these proteins is to interact with specific responsive elements on the DNA of genes. In recent years, miRNAs have been characterized as similar players at the mRNA level and thereby can regulate translation and mRNA turnover. As a result, different tissues are able to respond differently to the same stimulus such as interleukins, toxic compounds, drugs, and nutrients. This response is achieved by regulating gene expression and protein production. With regard to our study, we must hypothesize that the C3 gene in the kidney is regulated differently than the CRP gene in the liver. Moreover, the dynamics of CRP in serum is much brisker than that of C3.

At this point the question should be elaborated, why urinary C3a/C3 levels correlate negatively with serum CRP ($p < 0.0001$) in our study. The consumption and activation of C3 is a well-known marker for determining the activity of inflammatory diseases, such as rheumatic diseases or atypical hemolytic uremic syndrome. In this context, a decrease in C3 represents increased inflammation. Consequently, C3 can be said to represent an inverse acute phase parameter. In relation to our study, it is important to mention that mAb 3F7E2-based ELISA detects both C3 and C3a. To clarify the activation of C3 at the site of renal secretion, a commercially available C3a ELISA was performed that showed the presence of C3a in urine on all 6 days of measurement. The hypothesis that this measured C3a originates from the blood, because it is a small molecule and, unlike C3, could pass through the glomerular filter, must be rejected because C3a has a very short half-life and is rapidly degraded in the blood by proteases.

Overall, the question arises as to the purpose of this urinary C3 provided by the kidney. In this respect much earlier work by other authors has demonstrated that C3 transcription and protein production can be induced in a time and dose dependent fashion in proximal tubular epithelial cells by IL-1α [23]. In addition, other cytokines such as TNF-α, IL-6, IL-8, and MCP-1 are produced in response to IL-2 stimulation. It represents a matter of speculation if invading leukocytes or stressed epithelial cells themselves are the source of IL-1 and thereby induce C3 transcription and protein production. This then can act in both directions causing cell injury but at the same time preventing cell death through autophagy.

The shortcoming of this study is that C3a/C3 excretion could not be measured for the complete 6 days in all patients, as some died or were discharged before.

C3a/C3 is detected in urine upon infection-associated stimuli but declines rapidly following recovery from disease. It appears to represent an indicator of an acute phase reaction and stress causing condition, which in part might be involved in the regeneration of AKI-associated damage to epithelial cells, despite AKI-stages not being associated with urinary C3a/C3 levels in the measured timely interval.

## Supporting information

**S1 Raw images.**
(PDF)

## Acknowledgments

We would like to thank Univ.-Prof. Mag. Dr. Harald Heinzl from the Center for Medical Statistics, Informatics and Intelligent Systems of the Medical University of Vienna and Dr. Christian Gerges, PhD from the Division of Cardiology of the Department of Medicine II of the Medical University of Vienna for their support in statistical analysis.

## Author Contributions

**Conceptualization:** Sahra Pajenda, Ludwig Wagner, Daniela Gerges.

**Data curation:** Sahra Pajenda, Florence Zawedde, Sebastian Kapps, Ludwig Wagner.

**Formal analysis:** Sahra Pajenda, Sebastian Kapps, Ludwig Wagner, Daniela Gerges.

**Funding acquisition:** Ludwig Wagner, Alice Schmidt, Daniela Gerges.

**Investigation:** Sahra Pajenda, Florence Zawedde, Sebastian Kapps, Ludwig Wagner.

**Methodology:** Ludwig Wagner, David O'Connell.

**Project administration:** Ludwig Wagner, Wolfgang Winnicki, Daniela Gerges.

**Resources:** Ludwig Wagner, Alice Schmidt, Daniela Gerges.

**Supervision:** Ludwig Wagner, Daniela Gerges.

**Visualization:** Sahra Pajenda, Florence Zawedde, Sebastian Kapps, Ludwig Wagner, Wolfgang Winnicki, Daniela Gerges.

**Writing – original draft:** Sahra Pajenda, Ludwig Wagner.

**Writing – review & editing:** Sahra Pajenda, Ludwig Wagner, Alice Schmidt, Wolfgang Winnicki, David O'Connell, Daniela Gerges.

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
