## [Decision Letter · Decision Letter 0]

29 Mar 2021

PONE-D-21-05396

Are urinary C3 levels associated with the renal acute phase reaction in acute kidney disease? – A pilot study

PLOS ONE

Dear Dr. Gerges,

Thank you for submitting your manuscript to PLOS ONE. After careful consideration, we feel that it has merit but does not fully meet PLOS ONE’s publication criteria as it currently stands. Therefore, we invite you to submit a revised version of the manuscript that addresses the points raised during the review process.

Your manuscripts was reviewed by two experts and they are critical of the experimental design of the experiments. I am giving you an option for revision to address those technical issues.

We look forward to receiving your revised manuscript.

Kind regards,

Partha Mukhopadhyay, Ph.D.

Academic Editor

PLOS ONE

Journal Requirements:

Please compile all raw blot and gel images in a single PDF file titled S1_raw_images, and upload this file as Supporting Information or provide it via a publicly available data repository and include the dataset identifier (DOI or equivalent) in the Data Availability Statement. We ask that you ensure that every image in the file is clearly labeled to annotate the loading order, identity of experimental samples, method used to capture the image, and which figure panel was generated from that original image. Molecular weight markers should be included or indicated on each blot/gel image, and any lanes not included in the final figure should be marked with an “X” above the lane label on the original blot/gel image. All labeling and annotations should be performed without obscuring any data or background bands. Please note, there is a 20 MB maximum file size for Supporting Information files. If your PDF size is larger, please use a suitable repository or discuss with the journal staff.

In your cover letter, please note whether your blot/gel image data are in Supporting Information or posted at a public data repository, provide the repository URL if relevant, and provide specific details as to which raw blot/gel images, if any, are not available.

PLOS policy and the journal’s other requirements for blot/gel reporting and figure preparation are described in detail at https://journals.plos.org/plosone/s/figures#loc-blot-and-gel-reporting-requirements and https://journals.plos.org/plosone/s/figures#loc-preparing-figures-from-image-files. When you submit your revised manuscript, please ensure that your figures adhere fully to these guidelines and provide the original underlying images for all blot or gel data reported in your submission. See the following link for instructions on providing the original image data: https://journals.plos.org/plosone/s/figures#loc-original-images-for-blots-and-gels.

Furthermore, please specify in your ethics statement: 1) whether the ethics committee approved the verbal/oral consent procedure, 2) why written consent could not be obtained, and 3) how verbal/oral consent was recorded.

Reviewers' comments:

Reviewer's Responses to Questions

**Comments to the Author**

1. Is the manuscript technically sound, and do the data support the conclusions?

Reviewer #1: No

Reviewer #2: Partly

2. Has the statistical analysis been performed appropriately and rigorously? 

Reviewer #1: No

Reviewer #2: Yes

3. Have the authors made all data underlying the findings in their manuscript fully available?

Reviewer #1: Yes

Reviewer #2: Yes

4. Is the manuscript presented in an intelligible fashion and written in standard English?

Reviewer #1: Yes

Reviewer #2: No

5. Review Comments to the Author

Reviewer #1: The current study “Are urinary C3 levels associated with the renal acute phase reaction in acute kidney disease? – A pilot study” attempts to study the relationship of urinary C3 and acute phase renal reaction. However, there are many major problems in the experimental design.

1. The authors aim for the C3 molecules derived from the kidney. However, there is no indication the C3 molecules detected are from the kidney, instead of liver. In kidney injury, especially septic injury as discussed in the current work, it is very likely complement molecules leak through the blood vessels into the urine.

2. The authors used sepsis patients and analyzed the kidney injury and C3. However, C3 and kidney injury are both too much related to sepsis and any correlation shown in this study may be the result of their respective correlation with sepsis. This is a major statistical error in correlation study.

3. The authors found no correlation between AKI stage and C3 concentration. This means the study gives a negative result. However, the authors claims C3 molecules a good indicator. The conclusion contradicts with the data.

4. Urine concentration of a specific protein is very variable. So many factors interfere with the urine amount. Unless the authors provide a practical and standardized method, the study and its clinical relevance are not valid.

Reviewer #2: In the present study “Are urinary C3 levels associated with the renal acute phase reaction in acute kidney disease?-A pilot study” the authors try to discover the C3a/C3 as the adjunctive diagnostic marker for Acute kidney injury. The article might be helpful to understand urine C3a/C3 in the development of AKI. However, several concerns must be addressed for better quality. Here are some major concerns.

1. This study mainly rely on mAb 3F7E2 (C3a/C3 specific mouse monoclonal antibody) to detect C3a/C3. Why the specific Abs each of C3 and C3a were not used? Using of mAb 3F7E2 made it hard to distinguish whether the production of C3 or the activation of C3 is altered during the study.

2. Table 1 shows majority of patients had Pneumonia. Is there the possibility C3 from respiratory system transfer to Urinary system and affect the urine C3 level?

3. For Table 2, the finding from this study show Peak-uC3a/C3 is negatively correlated with CRP. Based on relevant literature, activation of complement factors are associated with increase inflammatory reaction. C3a levels or C3a to C3 ratio should be check to support authors’ idea.

4. For Fig1A, the authors claim that Dot blot initial screens test show decline in urinary C3a/C3 on Day 5 and 6. Please consider developing numeric scoring according to dot color and its C3a/C3 levels.

5. For Fig1B, degradation of C3 is mentioned in results. Though, C3a band seems not clear. Please indicate C3 and C3a location.

6. For Fig2, the indication should be changed to urinary C3a/C3. Some patients couldn’t make it until Day 5-6 with worsened condition. If C3/C3a is related with disease severity, it may affect decreased C3a/C3 level on Day 5-6.

7. For Fig3, please change the graph to dot form for the consistency.

8. Fig4 shows intracellular deposit of C3a/Ca in epithelial cells from urine sediment. Please include IF from control group without systemic inflammation or AKI.

6. PLOS authors have the option to publish the peer review history of their article (what does this mean?). If published, this will include your full peer review and any attached files.

Reviewer #1: No

Reviewer #2: No

---

## [Author Response · Author response to Decision Letter 0]

17 May 2021

Dear Editors in Chief,

Please find enclosed a revised version of our systematic review entitled Are urinary C3 levels associated with the renal acute phase reaction in acute kidney disease? – A pilot study for possible publication in PLOS ONE.

We thank the reviewers for their comments. Please find a point-by-point response letter to the reviewers’ comments below.

Thank you for considering our manuscript!

Yours sincerely,

Daniela Gerges, MD, MSc

 

Reviewer comments: black font

Response: blue font

Modifications in the revised version of the manuscript: red font

Major concerns:

Reviewer #1:

(C1) The authors aim for the C3 molecules derived from the kidney. However, there is no indication the C3 molecules detected are from the kidney, instead of liver. In kidney injury, especially septic injury as discussed in the current work, it is very likely complement molecules leak through the blood vessels into the urine.

(R1A): Thank you for this valuable and important comment as this point apparently needs further clarification in our work. Local synthesis of C3 has already been shown in the past by Sacks S et al. Sack S and colleagues were able to describe that glomerular mesangial cells are capable of producing C3 and C4, with an increase in C4 expression after stimulation with interferon-gamma, whereas C3 expression remains unaffected under interferon-gamma stimulation [1]. These data are substantiated by a further study, which demonstrates that C3 is synthesized, processed and secreted by glomerular epithelial cells under basal conditions, with the C3 alpha and beta polypeptide chains having identical electrophoretic mobilities with those of hepatic C3. In contrast to the study mentioned above, stimulation with interferon-gamma lead to an increase in C3 gene expression, indicating that C3 expression in glomerular epithelial cells is regulated by interferon-gamma [2]. In a further study using human samples of patients with postischemic acute renal failure, local synthesis of C3 in the tubule was described and the authors were able to distinguish tubular C3 from C3 of hepatic origin [3]. In addition, another study described enhanced local C3 production in immune-mediated nephritis [4].

Regarding our study, immunofluorescence staining clearly exhibits tubular epithelial cells containing intracellular C3 (Fig 7 in the manuscript). It is therefore hard to imagine how hepatic C3 could enter a tubular cell. It must be emphasized that the molecule size of 185 to 190 kDa is not suitable for passing through the glomerular filter. Although the slit diaphragm may be vulnerable in sepsis, such large molecules should not enter the urine in kidneys of sepsis patients. 

What appears very likely to occur, is that C3 is locally synthesized in tubular cells as has already been described in the literature. We included a statement regarding the origin of C3 in the discussion section of our manuscript.

Discussion, paragraph 2, page 13:

Local synthesis of C3 has already been described in the past. Sack S and colleagues were able to describe that glomerular mesangial cells are capable of producing C3 and C4, with an increase in C4 expression after stimulation with interferon-gamma (INF-gamma), whereas C3 expression remains unaffected under interferon-gamma stimulation [1]. These data are substantiated by a further study, which demonstrates that C3 is synthesized, processed, and secreted by glomerular epithelial cells under basal conditions, with the C3 alpha and beta polypeptide chains having identical electrophoretic mobilities with those of hepatic C3. In contrast to the study mentioned above, stimulation with INF-gamma lead to an increase in C3 gene expression, indicating that C3 expression in glomerular epithelial cells is regulated by INF-gamma [2]. In addition, increased local C3 synthesis has been described in human diseases, such as postischemic acute renal failure and immune-mediated nephritis [3, 4].

Figure 7 - manuscript

Fig 7. Immunofluorescence staining of cryosections of a kidney removed because of a renal cell carcinoma using rabbit anti human C3-specific antibody. Tubular epithelial cells contain intracellular C3 (left panel). Merging C3 staining with DAPI nuclear staining (right panel).

 

(R1B): To further substantiate and affirm our data, we performed a C3-specific RT-qPCR transcriptome analysis from the urinary sediment of AKI patients with and without kidney transplantation. This investigation was initiated in response to the comments of the reviewers.

These data were performed solely for the purpose of review and were not included into our manuscript as they provide data from a partially different cohort. This is because urine sediment was only available from a minor number of study participants:

Methods: 125 urine samples of a total of 64 patients were utilized. Twenty-four patients were kidney transplant recipients, one patient had received bone marrow transplantation and all patients experienced AKI. Urine sediment was palleted and RNA was extracted out of the sediment by mixing with Trizol. The Trizol lysate was then mixed with chlorophorm for precipitating the total RNA using isopropanol, as described in the Trizol test manual. Purified RNA was dissolved in RNAse free water and mixed with dNTPs random hexamer primers and reverse transcriptase using superscript enzyme. The resulting cDNA was diluted 1:4 with H2O and amplified using C3 specific probes from TaqMan® (Hs01100881_m1, Thermo Fisher Scientific) and 2x TaqMan® Universal Master Mix in a StepOnePlus qPCR machine (Applied Biosystems®). Data recording was performed over 46 cycles. Individual C3 expression levels in terms of cycle threshold (Ct) were normalized using GAPDH as house-keeping gene resulting in a ΔCt value, and further calculated using the ΔΔCt method [5]. Results: Forty-six samples showed no expression of C3 until 44 cycles and were therefore considered negative. Seventy-nine samples exhibited C3 transcripts in urine sediment cells or cell fragments, indicative for C3 synthesis.

Relative C3 expression normalized to GAPDH is given below in Reviewer material I.

 

Reviewer material I. Relative C3 expression in urinary sediments of AKI patients normalized to GAPDH. 125 urine samples of 64 patients were analyzed: 24 patients were kidney transplant recipients; one patient had received bone marrow transplantation and all patients experienced AKI. Each bar represents one patient. Values are given as relative expression of C3 normalized to GAPDH in urinary sediments.

 

(C2) The authors used sepsis patients and analyzed the kidney injury and C3. However, C3 and kidney injury are both too much related to sepsis and any correlation shown in this study may be the result of their respective correlation with sepsis. This is a major statistical error in correlation study.

(R2): As mentioned in R1, several other conditions are known, which lead to renal synthesis of C3, such as postischemic injury and immune mediated nephritis [3, 4]. However, we have included an analysis for the reviewer to validate our data and exhibit that also in contrast nephropathy C3/C3a excretion in urine is elevated one day after administration of contrast agent. It might be assumed that C3/C3a is upregulated in response to the adverse effect caused by the contrast agent or might exhibit cell-protective effects, such as autophagy. However, this is the content of work only performed for affirming our data and will not be included in the present study but are elucidated for the reviewer.

Reviewer material II. Urinary C3/C3a levels increase one day after intravenous CT contrast agent application. 45 patients were included in this work. Day 1 represents urinary C3/C3a levels prior to contrast agent application. Urinary C3/C3a levels increased on day 2, after application of contrast-CT and went down on day 3. Before contrast application all patients were checked for being negative for any inflammatory process.

  

(C3) The authors found no correlation between AKI stage and C3 concentration. This means the study gives a negative result. However, the authors claims C3 molecules a good indicator. The conclusion contradicts with the data.

(R3): Thank you for this valuable comment. Our study stated that C3a/C3 level correlated negatively with serum CRP, however, did not correlate with peak serum creatinine, peak leukocyte count and age. The individual patients’ inflammatory levels (i.e. CRP and urinary C3a/C3 levels) were independent of urinary output and glomerular filtration rate. We do not claim in this study, that urinary C3a/C3 levels are a good indicator for renal function (which would be of little to no clinical use), but rather an indicator for an acute phase reaction and cellular stress condition to the kidneys. We acknowledge that our statement can be misleading and therefore included in the Discussion section as follows:

Discussion, last paragraph , page 15:

C3a/C3 is detected in urine upon infection-associated stimuli but declines rapidly following recovery from disease. It appears to represent an indicator of an acute phase reaction and stress causing condition, which in part might be involved in the regeneration of AKI-associated damage to epithelial cells, despite AKI-stages not being associated with urinary C3a/C3 levels in the measured timely interval.

(C4) Urine concentration of a specific protein is very variable. So many factors interfere with the urine amount. Unless the authors provide a practical and standardized method, the study and its clinical relevance are not valid.

(R4): Thank you for this comment. The study was conducted as pilot study as mentioned in the title and does not claim to provide a novel, practical or standardized method for the measurement of urinary inflammatory parameters but rather give a new viewpoint into the understanding of inflammatory or stressing mechanisms involving the kidneys. Standardizing urinary C3a/C3 expression to urinary output or GFR would unfortunately miss the point of this study as inflammation is not necessarily associated with urinary output or renal function. Due to individual patients’ genetics and comorbidities standardizing urinary outputs to C3a/C3 levels would probably falsify the data. 

Reviewer #2:

(C1) This study mainly rely on mAb 3F7E2 (C3a/C3 specific mouse monoclonal antibody) to detect C3a/C3. Why the specific Abs each of C3 and C3a were not used? Using of mAb 3F7E2 made it hard to distinguish whether the production of C3 or the activation of C3 is altered during the study.

(R1): Thank you for this important comment. To estimate activation of C3 a C3a-ELISA was performed and methods and results included into this revision. 

Methods and materials, Page 7, paragraph 2:

Urine C3a ELISA

The human C3a ELISA (Invitrogen, BMS2089) was carried out as indicated in the test manual. Urine samples were thawed under airflow. The diluted (1:4 or 1:2 in sample diluent) urine samples (100µl) were pipetted into the prewashed test wells and incubated for two hours under constant shaking together with the provided standard series which was prepared in the sample diluent. Following washing on an automated ELISA washer (three times) the biotin-conjugate prepared in the assay buffer (100µl) was pipetted into each test well and incubated for one hour at RT under constant shaking. Following a second washing step again three times, the streptavidin-HRP prepared in the assay buffer provided with the test kit (100µl) was added into each well and incubated for one hour under constant shaking at RT. After a final washing step, the TMB substrate solution (100µl) was pipetted into each well and reacted at RT for 20 minutes under light protection. The reaction was then stopped by adding 100µl stop solution. The test signal was read at 450nm at an ELISA reader and the sample concentrations were calculated according to the standard curve using the Gen5 version 2.03 program.

Results, Page 9-10, subheading ELISA testing:

As depicted in the left panel of Fig 2 the highest mean overall level of C3a/C3 was found on the second day of measurement, which represented the second day of treatment (all p<0.05). Selectively measured C3a was detectable in urine over all 6 days (Fig 2 right panel). Out of 230 samples 178 exhibited detectable levels of C3a as shown in (Fig 2 right panel).

Page 14-15, last and first paragraph:

For that reason, we performed a C3a specific ELISA measurement, but C3a represents a small molecule capable of passing the glomerular filter.

 

Figure 2 manuscript:

Fig 2. ELISA-measurement of urinary complement factor C3a/C3 and C3a in 85 ICU patients. Data are presented in columned scatter graphs; the horizontal lines mark the mean values of the respective C3a/C3 (left) and C3a (right) levels. Day 1: n=78; day 2: n=78; day 3: n=72; day 4: n=66; day 5: n=56; day 6: n=48. 

 

(C2) Table 1 shows majority of patients had Pneumonia. Is there the possibility C3 from respiratory system transfer to Urinary system and affect the urine C3 level?

(R2): Thank you for this important comment. It is very unlikely that C3 from the respiratory system was transferred to the urinary tract. In fact, local synthesis of C3 in renal cells has been described in the past. We therefore added a statement to the discussion, giving already published data of local C3 synthesis by former authors.

Discussion, paragraph 2, page 13:

Local synthesis of C3 has already been described in the past. Sack S and colleagues were able to describe that glomerular mesangial cells are capable of producing C3 and C4, with an increase in C4 expression after stimulation with interferon-gamma (INF-gamma), whereas C3 expression remains unaffected under interferon-gamma stimulation [1]. These data are substantiated by a further study, which demonstrates that C3 is synthesized, processed and secreted by glomerular epithelial cells under basal conditions, with the C3 alpha and beta polypeptide chains having identical electrophoretic mobilities with those of hepatic C3. In contrast to the study mentioned above, stimulation with INF-gamma lead to an increase in C3 gene expression, indicating that C3 expression in glomerular epithelial cells is regulated by INF-gamma [2]. In addition, increased local C3 synthesis has been described in human diseases, such as postischemic acute renal failure and immune-mediated nephritis [3, 4].

(C3) For Table 2, the finding from this study show Peak-uC3a/C3 is negatively correlated with CRP. Based on relevant literature, activation of complement factors are associated with increase inflammatory reaction. C3a levels or C3a to C3 ratio should be check to support authors’ idea.

(R3): Due to the fact that the mAb 3F7E2 based ELISA detects both C3a and C3 no ratio of C3a and C3 could be calculated. However, we performed following the suggestion of the reviewer a commercially available C3a specific ELISA and provide urine concentration levels of C3a levels in an additional graph (Fig. 2 right panel).

 

(C4) For Fig1, the authors claim that Dot blot initial screens test show decline in urinary C3a/C3 on Day 5 and 6. Please consider developing numeric scoring according to dot color and its C3a/C3 levels.

(R4): Thank you for this important comment. Values were assigned and the following table was added to the manuscript.

Methods section, subheading Dot plot, page 6:

Dot blot densitometric evaluation was performed using FusionCapt Advance Solo 4 16.06.

Table 1, page 17:

Table 1. Densitometric values of C3/C3a dot blot analysis of 20 AKI patients at the ICU. Urine samples were obtained at consecutive time points over 6 days.

ID Day 1 Day 2 Day 3 Day 4 Day 5 Day 6

a 307 260 355 411 100 91

b 56 98 165 79 97 38

c 167 154 223 52 61 na

d 36 44 222 74 38 na

e 105 97 74 41 39 58

f 34 50 158 169 109 na

g 40 78 93 50 184 na

h 294 na 239 95 na na

i 164 70 130 107 36 47

j 248 239 370 198 85 na

k 48 207 78 63 76 na

l 104 138 115 68 90 77

m 372 327 162 168 249 na

n 161 238 123 69 na na

o 112 113 85 297 724 na

p 108 49 52 59 113 200

q 51 134 140 45 97 144

r na na 54 40 141 267

s na 302 197 116 686 490

t 186 150 398 327 436 438

na=not applicable.

(C5) For Fig 4, degradation of C3 is mentioned in results. Though, C3a band seems not clear. Please indicate C3 and C3a location.

(R5): Thank you for this important comment. Fig 4 does show C3 only, C3a is not shown. This was clarified in the Figure caption. Also, the sentence regarding the degradation was corrected.

Figure 4 - manuscript:

Fig 4. C3 serum und urine immunoblot using mAb 3F7E2. Ten microlitres of human urine (patient#1 and #2) was loaded onto a 10% SDS-PAGE transferred to nitrocellulose and incubated with mAb 3F7E2 (right panel). 0.4µl of human serum (patient#1 and #2) was loaded onto a 10% SDS-PAGE transferred to nitrocellulose and incubated with mAb 3F7E2 (middle panel). Molecular weight marker is shown at the left panel. Shown is a representative experiment out of two. The entire 190 kDa C3 molecule was detected. C3a could not be delineated. 

(C6) For Fig2, the indication should be changed to urinary C3a/C3. Some patients couldn’t make it until Day 5-6 with worsened condition. If C3/C3a is related with disease severity, it may affect decreased C3a/C3 level on Day 5-6.

(R6): Thank you for this important comment. The indication was changed accordingly (see C1 – Figure 2 manuscript). 

 

(C7) For Fig3, please change the graph to dot form for the consistency.

(R7): Thank you for this valuable comment. We have changed the graph.

Figure 3 manuscript:

Fig 3. Peak urinary C3a/C3 level in patients categorized for stage of AKI. Peak levels of urinary C3a/C3 did not correlate with AKI stage. AKI 0 (n=21), AKI 1 (n=37), AKI 2 (n=10) AKI 3 (n=17).

 

(C8) Fig 6 shows intracellular deposit of C3a/C3 in epithelial cells from urine sediment. Please include IF from control group without systemic inflammation or AKI.

(R8): Thank you for this valuable comment. We included a figure with IF of a control patient without systemic inflammation or AKI and included it into Figure 6.

Results, subheading Immunofluorescence, page 11:

A negative control from a patient with non-inflammatory condition is included in Fig 6D.

Figure 6:

Fig 6. Immunofluorescence staining of urinary sediment using mAb 3F7E2. Tubular epithelial cells contain intracellular granules with C3 (A, B). Bacteria present in the urinary sediment are veiled in C3a/C3 to some extent (C). The scale bar represents 10 µm. Immunofluorescence staining of a urinary sediment of a patient without systemic inflammation and AKI served as control (D). The scale bar represents 10 µm.

 

References

1. Sacks S, Zhou W, Campbell RD, Martin J. C3 and C4 gene expression and interferon-gamma-mediated regulation in human glomerular mesangial cells. Clin Exp Immunol. 1993;93(3):411-7. Epub 1993/09/01. PubMed PMID: 8370168; PubMed Central PMCID: PMC1554924.

2. Sacks SH, Zhou W, Pani A, Campbell RD, Martin J. Complement C3 gene expression and regulation in human glomerular epithelial cells. Immunology. 1993;79(3):348-54. Epub 1993/07/01. PubMed PMID: 8406564; PubMed Central PMCID: PMC1421987.

3. Farrar CA, Zhou W, Lin T, Sacks SH. Local extravascular pool of C3 is a determinant of postischemic acute renal failure. Faseb J. 2006;20(2):217-26. Epub 2006/02/02. doi: 10.1096/fj.05-4747com. PubMed PMID: 16449793.

4. Sacks SH, Zhou W, Andrews PA, Hartley B. Endogenous complement C3 synthesis in immune complex nephritis. Lancet. 1993;342(8882):1273-4. Epub 1993/11/20. PubMed PMID: 7901586.

5. Livak KJ, Schmittgen TD. Analysis of relative gene expression data using real-time quantitative PCR and the 2(-Delta Delta C(T)) Method. Methods. 2001;25(4):402-8. Epub 2002/02/16. doi: 10.1006/meth.2001.1262. PubMed PMID: 11846609.

NOTE: Please note, that Figure numbers were updated, as they appear in the latest version of the manuscript.

---

## [Decision Letter · Decision Letter 1]

18 Jun 2021

PONE-D-21-05396R1

Are urinary C3 levels associated with the renal acute phase reaction in acute kidney disease? – A pilot study

PLOS ONE

Dear Dr. Gerges,

Thank you for submitting your manuscript to PLOS ONE. After careful consideration, we have decided that your manuscript does not meet our criteria for publication and must therefore be rejected.

Specifically:

Your revised manuscript did not address main technical concerns and the data is sufficient to support the conclusions. However, the manuscript is interesting and  you may submit it as "new submission" after addressing all major technical aspects.

I am sorry that we cannot be more positive on this occasion, but hope that you appreciate the reasons for this decision.

Yours sincerely,

Partha Mukhopadhyay, Ph.D.

Academic Editor

PLOS ONE

Reviewers' comments:

Reviewer's Responses to Questions

**Comments to the Author**

1. If the authors have adequately addressed your comments raised in a previous round of review and you feel that this manuscript is now acceptable for publication, you may indicate that here to bypass the “Comments to the Author” section, enter your conflict of interest statement in the “Confidential to Editor” section, and submit your "Accept" recommendation.

Reviewer #1: (No Response)

Reviewer #2: All comments have been addressed

2. Is the manuscript technically sound, and do the data support the conclusions?

Reviewer #1: No

Reviewer #2: (No Response)

3. Has the statistical analysis been performed appropriately and rigorously? 

Reviewer #1: No

Reviewer #2: (No Response)

4. Have the authors made all data underlying the findings in their manuscript fully available?

Reviewer #1: Yes

Reviewer #2: (No Response)

5. Is the manuscript presented in an intelligible fashion and written in standard English?

Reviewer #1: Yes

Reviewer #2: (No Response)

6. Review Comments to the Author

Reviewer #1: The current version of “Are urinary C3 levels associated with the renal acute phase reaction in acute kidney disease? – A pilot study” did not answer the following critical concerns. The results are not sufficient to support the conclusion of the study.

1. The authors cannot trace the source of urine C3. Even though other studies and the authors found an expression of C3 from kidney cells, it is still very likely that most of the C3 found in the urine is from the liver. The authors proposed that the C3 is too high in molecular weight so that it cannot pass the filtration barrier. However, immunoglobulins are frequently found to pass through the same barrier in kidney conditions and they have similar molecular weight. These claims are thus not convincing.

2. The authors did not exclude the possibility that the C3 production and the kidney damage are both independently correlated to the sepsis, while C3 production/leakage and kidney damage are not linked in the current study.

3. The authors claimed that the urinary C3 an “indicator for an acute phase reaction and cellular stress condition to the kidneys”. However, according to the experimental method, the authors did not trace the course of disease to define an acute phase. Refining an acute phase of sepsis is very difficult to achieve in practice. Data show that C3 and CRP concentration are negatively correlated, which is the opposite of the conclusion, as CRP is one of the best indicators of acute phase reaction. The authors did not examine cellular stress of the kidney either. These conclusions are not supported by the data.

Reviewer #2: (No Response)

7. PLOS authors have the option to publish the peer review history of their article (what does this mean?). If published, this will include your full peer review and any attached files.

Reviewer #1: No

Reviewer #2: No

- - - - -

---

## [Author Response · Author response to Decision Letter 1]

12 Jul 2021

(C1) The authors cannot trace the source of urine C3. Even though other studies and the authors found an expression of C3 from kidney cells, it is still very likely that most of the C3 found in the urine is from the liver. The authors proposed that the C3 is too high in molecular weight so that it cannot pass the filtration barrier. However, immunoglobulins are frequently found to pass through the same barrier in kidney conditions and they have similar molecular weight. These claims are thus not convincing.

(R1A): We hereby attempt to defend and clarify our experience and position, although we understand and have included the reviewer's statements as strong arguments: As mentioned before, local synthesis of C3 has already been shown in the past by Sacks S et al. Sack S et al described the capability of glomerular mesangial cells to produce C3 and C4, with an increase in C4 expression after stimulation with interferon-gamma, whereas C3 expression remains unaffected under interferon-gamma stimulation [1]. These data are substantiated by a further study, which demonstrates that C3 is synthesized, processed, and secreted by glomerular epithelial cells under basal conditions, with the C3 alpha and beta polypeptide chains having identical electrophoretic mobilities with those of hepatic C3. In contrast to the study mentioned above, stimulation with interferon-gamma lead to an increase in C3 gene expression, indicating that C3 expression in glomerular epithelial cells is regulated by interferon-gamma [2]. In a further study using human samples of patients with postischemic acute renal failure, local synthesis of C3 in the tubule was described and the authors were able to distinguish tubular C3 from C3 of hepatic origin [3]. In addition, another study described enhanced local C3 production in immune-mediated nephritis [4]. 

IgG is ascribed a size of 150 kDa, whereas C3 is estimated to be significantly larger at 190 kDa; moreover, the molecular structure of IgG is quite different from that of C3. This argues against the reviewer's hypothesis that IgG excretion can be taken as a comparative molecule to C3 in its amount of excretion through the slit diaphragm of the glomerulus in sepsis.

In addition, in all individuals included in the Human Protein Atlas, which is open access, C3 is expressed to variable extent such as shown below. The extracted figure below represents the individual CAB004209, Female, age 56 and was stained for C3 by a rabbit monospecific antibody. 

Reviewer material I. This figure has not been produced by our research group. It represents one extract from the human protein atlas: https://www.proteinatlas.org/ENSG00000125730-C3/tissue/kidney#img. Further images can be seen obtained each from another individual even in children age 7, but all express C3.

Concerning our study, immunofluorescence staining clearly demonstrates tubular epithelial cells containing intracellular C3 (Fig. 7 in the manuscript). 

 

Figure 7 from main manuscript

Fig 7. Immunofluorescence staining of cryosections of a kidney removed because of a renal cell carcinoma using rabbit anti human C3-specific antibody. Tubular epithelial cells contain intracellular C3 (left panel). Merging C3 staining with DAPI nuclear staining (right panel).

Two modes of secretion are known in biology. It has to be assumed that C3 is not constitutively secreted into urine. There must be a regulated mode of secretion from nephronic epithelial cells. A specific stimulus secretion coupling has to exist. One of them might occur in infection and sepsis. The exact triggering molecule is not characterized but it must differ from that in the liver.

To further support and confirm our data, we performed C3-specific RT-qPCR transcriptome analysis from urine sediments of AKI patients with and without kidney transplantation. 

This study was initiated in response to reviewer comments. These data were performed for review purposes only and were not included in our manuscript because they provide data from a partially different cohort. This is because urine sediment was only available from a small number of study participants. In this analysis, forty-six samples showed no expression of C3 by cycle 44 and were therefore considered negative. Seventy-nine samples had C3 transcripts in the cells or cell fragments of the urine sediment, indicating C3 synthesis.

 

Methods: 125 urine samples of a total of 64 patients were utilized. Twenty-four patients were kidney transplant recipients, one patient had received bone marrow transplantation and all patients experienced AKI. Urine sediment was palleted and RNA was extracted out of the sediment by mixing with Trizol. The Trizol lysate was then mixed with chlorophorm for precipitating the total RNA using isopropanol, as described in the Trizol test manual. Purified RNA was dissolved in RNAse free water and mixed with dNTPs random hexamer primers and reverse transcriptase using superscript enzyme. The resulting cDNA was diluted 1:4 with H2O and amplified using C3 specific probes from TaqMan® (Hs01100881_m1, Thermo Fisher Scientific) and 2x TaqMan® Universal Master Mix in a StepOnePlus qPCR machine (Applied Biosystems®). Data recording was performed over 46 cycles. Individual C3 expression levels in terms of cycle threshold (Ct) were normalized using GAPDH as house-keeping gene resulting in a ΔCt value, and further calculated using the ΔΔCt method [5]. 

Results: Forty-six samples showed no expression of C3 until 44 cycles and were therefore considered negative. Seventy-nine samples exhibited C3 transcripts in urine sediment cells or cell fragments, indicative for C3 synthesis. Relative C3 expression normalized to GAPDH is given below in Reviewer material II.

Reviewer material II. Relative C3 expression in urinary sediments of AKI patients normalized to GAPDH. 125 urine samples of 64 patients were analyzed: 24 patients were kidney transplant recipients; one patient had received bone marrow transplantation and all patients experienced AKI. Each bar represents one patient. Values are given as relative expression of C3 normalized to GAPDH in urinary sediments.

Furthermore, we have again included a further analysis only performed for the purpose of review, which shall serve to validate our data. It exhibits that also in contrast nephropathy excretion of C3/C3a in urine is increased one day after administration of contrast agent. Therefore, it might be assumed that C3/C3a is upregulated or the secretion out of the tubular cells is stimulated in response to the adverse effect caused by the contrast agent or might exhibit cell-protective effects, such as autophagy. At this point it must be mentioned that none of these patients exhibited signs of infection and did not show elevated CRP levels. These data are nevertheless content of another study and are shown here for the sole purpose of review to again substantiate our data and will not be included in the present study but are elucidated for the reviewer.

Reviewer material III. Urinary C3/C3a levels increase one day after intravenous CT contrast agent application. 45 patients were included in this work. Day 1 represents urinary C3/C3a levels prior to contrast agent application. Urinary C3/C3a levels increased on day 2, after application of contrast-CT and went down on day 3. Before contrast application all patients were checked for being negative for any inflammatory process.

What seems very probable is that C3 is synthesized locally in tubular cells, as has been described in prior literature. We have included a statement about the origin of C3 with the available literature to our discussion section within the previous revision. 

Page 13, paragraph 1, Discussion, 

Local synthesis of C3 has already been described in the past. Sack S and colleagues were able to describe that glomerular mesangial cells are capable of producing C3 and C4, with an increase in C4 expression after stimulation with INF𝛾, whereas C3 expression remains unaffected under interferon-gamma stimulation [1]. These data are substantiated by a further study, which demonstrates that C3 is synthesized, processed, and secreted by glomerular epithelial cells under basal conditions, with the C3 alpha and beta polypeptide chains having identical electrophoretic mobilities with those of hepatic C3. In contrast to the study mentioned above, stimulation with INF𝛾 lead to an increase in C3 gene expression, indicating that C3 expression in glomerular epithelial cells is regulated by INF𝛾 [2]. In addition, increased local C3 synthesis has been described in human diseases, such as postischemic acute renal failure and immune-mediated nephritis [3, 4].

(C2) The authors did not exclude the possibility that the C3 production and the kidney damage are both independently correlated to the sepsis, while C3 production/leakage and kidney damage are not linked in the current study.

(R2) Transcription factor and repressor proteins are expressed in a tissue- and cell subtype-specific manner. For example, renal epithelial cells express different transcription factors and repressor proteins than hepatocytes. The function of these proteins is to interact with specific responsive elements on the DNA of genes. In recent years, miRNAs have been characterized as similar players at the mRNA level and thereby can regulate translation and mRNA turnover. As a result, different tissues are able to respond differently to the same stimulus such as interleukins, toxic compounds, drugs, and nutrients. This response is achieved by regulating gene expression and protein production. With regard to our study, we must hypothesize that the C3 gene in the kidney is regulated differently than the CRP gene in the liver. Moreover, the dynamics of CRP in serum is much more brisk than that of C3.  

(C3) The authors claimed that the urinary C3 an “indicator for an acute phase reaction and cellular stress condition to the kidneys”. However, according to the experimental method, the authors did not trace the course of disease to define an acute phase. Refining an acute phase of sepsis is very difficult to achieve in practice. Data show that C3 and CRP concentration are negatively correlated, which is the opposite of the conclusion, as CRP is one of the best indicators of acute phase reaction. The authors did not examine cellular stress of the kidney either. These conclusions are not supported by the data.

(R3): As described in the Methods section under the subheading “sample collection” only patients admitted to the intensive care unit (ICU) fulfilling at least 4 of 10 sepsis criteria were included into this study [6]. The sepsis criteria were established to define an acute phase of sepsis and are therefore the adequate tool to substantiate the fact that our patients were acutely ill. It is true, that CRP and urinary C3a/C3 levels correlated negatively with serum CRP (p<0.0001), however the mAb 3F7E2 based ELISA detects both, C3 and C3a. Consumption and activation of C3 is a known marker for determining the activity of inflammatory diseases, such as rheumatic diseases or atypical hemolytic uremic syndrome. Thereby, a decrease of C3 stands for increased inflammation. Therefore, it can be said, that for example like serum albumin, C3 represents an inverse acute phase parameter.

It is true that the CRP level is the best indicator of an acute-phase response, which is orchestrated by the liver and represents the response to a pathological stimulus. However, we disagree with the reviewer's assumption that the CRP level may be one of the best indicators of an acute-phase response in the kidney. The site in the kidney directly affected by sepsis is the endothelium and signaling to the tubular epithelial cells and podocytes occurs with a time delay. The tubular epithelial cells undergo specific response processes involving IFN signaling and STAT1. The kidney has different modalities to respond, often delayed, such as upregulation of alpha-2-macroglobulin, while CRP is not found elevated or not elevated in the blood in such disease [7]. 

An attempt to follow the reviewer's suggestion to better represent the acute phase reaction in the kidney, we wanted to elucidate the activation of C3 at the site of renal secretion, a commercially available C3a ELISA was performed in response to an acute phase reaction, and the data have already been included in the previous review. Again, the reviewer might say that this is from blood, since C3a is a small molecule. This is contradicted by the assumption that C3a has a very short half-life and is rapidly degraded by proteases in the blood.  

Methods, page 7, paragraph 2

Urine C3a ELISA

The human C3a ELISA (Invitrogen, BMS2089) was carried out as indicated in the test manual. Urine samples were thawed under airflow. The diluted (1:4 or 1:2 in sample diluent) urine samples (100µl) were pipetted into the prewashed test wells and incubated for two hours under constant shaking together with the provided standard series which was prepared in the sample diluent. Following washing on an automated ELISA washer (three times) the biotin-conjugate prepared in the assay buffer (100µl) was pipetted into each test well and incubated for one hour at RT under constant shaking. Following a second washing step again three times, the streptavidin-HRP prepared in the assay buffer provided with the test kit (100µl) was added into each well and incubated for one hour under constant shaking at RT. After a final washing step, the TMB substrate solution (100µl) was pipetted into each well and reacted at RT for 20 minutes under light protection. The reaction was then stopped by adding 100µl stop solution. The test signal was read at 450nm at an ELISA reader and the sample concentrations were calculated according to the standard curve using the Gen5 version 2.03 program.

Results, page 9-10, paragraph 3

As depicted in the left panel of Fig 2 the highest mean overall level of C3a/C3 was found on the second day of measurement, which represented the second day of treatment (all p<0.05). Selectively measured C3a was detectable in urine over all 6 days (Fig 2 right panel). Out of 230 samples 178 exhibited detectable levels of C3a as shown in (Fig 2 right panel).

Fig 2. ELISA-measurement of urinary complement factor C3a/C3 and C3a in 85 ICU patients. Data are presented in columned scatter graphs; the horizontal lines mark the mean values of the respective C3a/C3 (left) and C3a (right) levels. Day 1: n=78; day 2: n=78; day 3: n=72; day 4: n=66; day 5: n=56; day 6: n=48. 

  

References

1. Sacks S, Zhou W, Campbell RD, Martin J. C3 and C4 gene expression and interferon-gamma-mediated regulation in human glomerular mesangial cells. Clin Exp Immunol. 1993;93(3):411-7. Epub 1993/09/01. PubMed PMID: 8370168; PubMed Central PMCID: PMC1554924.

2. Sacks SH, Zhou W, Pani A, Campbell RD, Martin J. Complement C3 gene expression and regulation in human glomerular epithelial cells. Immunology. 1993;79(3):348-54. Epub 1993/07/01. PubMed PMID: 8406564; PubMed Central PMCID: PMC1421987.

3. Farrar CA, Zhou W, Lin T, Sacks SH. Local extravascular pool of C3 is a determinant of postischemic acute renal failure. Faseb J. 2006;20(2):217-26. Epub 2006/02/02. doi: 10.1096/fj.05-4747com. PubMed PMID: 16449793.

4. Sacks SH, Zhou W, Andrews PA, Hartley B. Endogenous complement C3 synthesis in immune complex nephritis. Lancet. 1993;342(8882):1273-4. Epub 1993/11/20. PubMed PMID: 7901586.

5. Livak KJ, Schmittgen TD. Analysis of relative gene expression data using real-time quantitative PCR and the 2(-Delta Delta C(T)) Method. Methods. 2001;25(4):402-8. Epub 2002/02/16. doi: 10.1006/meth.2001.1262. PubMed PMID: 11846609.

6. Funk D, Sebat F, Kumar A. A systems approach to the early recognition and rapid administration of best practice therapy in sepsis and septic shock. Curr Opin Crit Care. 2009;15(4):301-7. Epub 2009/06/30. doi: 10.1097/MCC.0b013e32832e3825. PubMed PMID: 19561493.

7. Menon R, Otto EA, Hoover P, Eddy S, Mariani L, Godfrey B, et al. Single cell transcriptomics identifies focal segmental glomerulosclerosis remission endothelial biomarker. JCI Insight. 2020;5(6). Epub 2020/02/29. doi: 10.1172/jci.insight.133267. PubMed PMID: 32107344; PubMed Central PMCID: PMCPMC7213795.

---

## [Decision Letter · Decision Letter 2]

27 Oct 2021

Urinary C3 levels associated with sepsis and acute kidney injury – A pilot study

PONE-D-21-05396R2

Dear Dr. Gerges,

We’re pleased to inform you that your manuscript has been judged scientifically suitable for publication and will be formally accepted for publication once it meets all outstanding technical requirements.

Kind regards,

Partha Mukhopadhyay, Ph.D.

Section Editor

PLOS ONE

Additional Editor Comments (optional):

Reviewers' comments:

Reviewer's Responses to Questions

**Comments to the Author**

1. If the authors have adequately addressed your comments raised in a previous round of review and you feel that this manuscript is now acceptable for publication, you may indicate that here to bypass the “Comments to the Author” section, enter your conflict of interest statement in the “Confidential to Editor” section, and submit your "Accept" recommendation.

Reviewer #3: (No Response)

2. Is the manuscript technically sound, and do the data support the conclusions?

Reviewer #3: Yes

3. Has the statistical analysis been performed appropriately and rigorously? 

Reviewer #3: Yes

4. Have the authors made all data underlying the findings in their manuscript fully available?

Reviewer #3: Yes

5. Is the manuscript presented in an intelligible fashion and written in standard English?

Reviewer #3: Yes

6. Review Comments to the Author

Reviewer #3: (No Response)

7. PLOS authors have the option to publish the peer review history of their article (what does this mean?). If published, this will include your full peer review and any attached files.

Reviewer #3: No

---

## [Editor Report · Acceptance letter]

29 Oct 2021

PONE-D-21-05396R2 

Urinary C3 levels associated with sepsis and acute kidney injury – A pilot study 

Dear Dr. Gerges:

I'm pleased to inform you that your manuscript has been deemed suitable for publication in PLOS ONE. Congratulations! Your manuscript is now with our production department. 

Kind regards, 

on behalf of

Dr. Partha Mukhopadhyay 

Section Editor

PLOS ONE